# CM-YOLOv8: Lightweight YOLO for Coal Mine Fully Mechanized Mining Face

**DOI:** 10.3390/s24061866

**Published:** 2024-03-14

**Authors:** Yingbo Fan, Shanjun Mao, Mei Li, Zheng Wu, Jitong Kang

**Affiliations:** Institute of Remote Sensing and Geographic Information Systems, Peking University, No. 5 Summer Palace Road, Beijing 100871, China; ybfan@stu.pku.edu.cn (Y.F.); mli@pku.edu.cn (M.L.); zheng_wu@pku.edu.cn (Z.W.); 2101210061@stu.pku.edu.cn (J.K.)

**Keywords:** object detection, coal mine, YOLOv8, lightweight pruning

## Abstract

With the continuous development of deep learning, the application of object detection based on deep neural networks in the coal mine has been expanding. Simultaneously, as the production applications demand higher recognition accuracy, most research chooses to enlarge the depth and parameters of the network to improve accuracy. However, due to the limited computing resources in the coal mining face, it is challenging to meet the computation demands of a large number of hardware resources. Therefore, this paper proposes a lightweight object detection algorithm designed specifically for the coal mining face, referred to as CM-YOLOv8. The algorithm introduces adaptive predefined anchor boxes tailored to the coal mining face dataset to enhance the detection performance of various targets. Simultaneously, a pruning method based on the L1 norm is designed, significantly compressing the model’s computation and parameter volume without compromising accuracy. The proposed algorithm is validated on the coal mining dataset DsLMF+, achieving a compression rate of 40% on the model volume with less than a 1% drop in accuracy. Comparative analysis with other existing algorithms demonstrates its efficiency and practicality in coal mining scenarios. The experiments confirm that CM-YOLOv8 significantly reduces the model’s computational requirements and volume while maintaining high accuracy.

## 1. Introduction

Today, with the continuous growth of global energy demand, coal, as a crucial energy resource that has served industrial production for centuries, still plays a vital role in the modern energy system [1,2,3]. It is currently widely used in electricity generation and industrial manufacturing, providing reliable energy support for societal development [4,5,6]. However, despite the undeniable significance of coal in the energy industry, the process of coal mining still faces a series of technical difficulties and challenges. Coal mining methods are typically associated with highly hazardous working environments, such as the risks of gas explosions, roof collapses, and other safety hazards, posing a threat to the life and safety of miners [7,8]. To enhance the safety of coal mining, it is necessary to introduce advanced safety identification technologies and systems to achieve more effective monitoring and prevention. Simultaneously, the widespread application of comprehensive intelligent mining technologies has also brought about new challenges [9,10]. In these highly automated work areas, the rapid and precise identification of various production processes and surrounding environments has become a pivotal factor in ensuring production efficiency. Traditional methods of manual observation and monitoring are inadequate to meet the demands of modern coal mining. Therefore, there is a need to rely on advanced image recognition and computer vision technologies to achieve real-time monitoring of the working face’s status [11,12].

Currently, common image recognition methods for coal mine fully mechanized working faces mainly include traditional methods based on feature extraction and deep-learning-based methods. Traditional approaches employ image-processing techniques such as edge detection, texture analysis, and shape descriptors. These techniques extract manually designed features from images, which are subsequently utilized for target detection and classification [13]. Presently, machine-learning-based methods, such as SVM (Support Vector Machine) [14] or Random Forest [15], are more widely applied. These methods combine manually extracted features for the classification and detection of coal mine images. Alternatively, Convolutional Neural Networks (CNNs), such as LeNet [16], AlexNet [17], VGG [18], or deeper networks are employed for end-to-end feature learning and target detection of images from fully mechanized coal mine working faces [19]. Further advancements in deep-learning-based object detection algorithms, such as Faster R-CNN [20], YOLO (You Only Look Once) [21], SSD (Single Shot MultiBox Detector) [22], also enable an efficient and precise localization and classification of targets in coal mine images.

However, for the paramount safety requirements of coal mine working faces, real-time and rapid identification of abnormal situations to minimize accident risks is an extremely crucial demand [23]. The timely detection of geological structural changes, roof collapses, hazardous personnel movements, or gas leaks is essential for taking prompt measures [24]. Therefore, in coal mine working faces, real-time capability is a key requirement, particularly for automation and safety monitoring systems [25]. Some deep learning algorithms may face speed challenges due to computational resource limitations. Additionally, coal mine working faces are often constrained by computational resources, necessitating lightweight algorithms for rapid image processing and recognition in resource-constrained environments [26].

To address the aforementioned issues, this paper proposes a lightweight object detection algorithm tailored for fully mechanized coal mine working faces. The algorithm is based on the YOLOv8 [27] object detection algorithm. Considering the typically fixed sizes and proportions of various identifiable targets within coal mine working face scenes, this paper designs predefined Anchor Boxes to predict the sizes of target boxes. Simultaneously, a lightweight optimization is applied to the network structure of YOLOv8. This optimization involves pruning operations based on the L1 norm for most CBS (Convolution, Batch Normalization, and Leaky ReLU) convolution modules [28] and some convolution operations in the network structure. This ensures that the object detection algorithm achieves more efficient hardware deployment and faster recognition speed when facing scenarios like fully mechanized coal mine working faces for target identification. In the design of predefined Anchor Boxes, this paper initially analyzes the dataset of fully mechanized coal mine working faces, gathering information on target width, height, and other aspects. Subsequently, the K-means clustering algorithm [29] is employed to cluster the target boxes in the dataset, thereby determining the sizes of the predefined Anchor Boxes. During clustering, the width and height of the target boxes can be utilized as features.

After performing K-means clustering, manual adjustments to the sizes of Anchor Boxes can be made to ensure their suitability for the shapes and proportions of targets in fully mechanized coal mine working faces. If there are changes in the distribution of targets in the working faces, the clustering algorithm can be periodically rerun to adapt to the new data distribution, ensuring that the Anchor Boxes can still effectively capture the sizes of the targets. In designing pruning and quantization strategies, this paper adopts L1 norm-based pruning operations for the majority of convolution kernels. The pruned portions are convolved with zero-filled kernels. Additionally, different compression ratios can be dynamically adjusted according to the specific requirements of different practical scenarios, accommodating the actual needs of fully mechanized coal mine working faces. This approach supports the efficient operation of the object detection algorithm on resource-constrained embedded systems and mobile devices.

## 2. Related Work

In the field of computer vision, object detection is a fundamental task that plays a crucial role in various human life and production applications, including areas such as autonomous driving, robotics, and intelligent security [30]. It has evolved from the extraction of handcrafted local invariant features, such as SIFT (Scale-Invariant Feature Transform) [31], HOG (Histogram of Oriented Gradients) [32], and LBP (Local Binary Patterns) [33]. The process then involves the aggregation of local features, achieved through simple cascading or encoders, such as the SPM (Spatial Pyramid Matching) [34] and Fisher Vectors [35]. For many years, the dominant paradigm in computer vision relied on handcrafted local descriptors and discriminative classifiers, forming a multi-stage process. The landscape changed with the unprecedented success of DCNNs (Deep Convolutional Neural Networks) in image classification [36]. The success of DCNNs in image classification propelled a paradigm shift that extended into the field of object detection [37].

Until recent years, the representative two-stage object detector, RCNN, was proposed. It initially extracts candidate boxes based on the image and then refines the detection results by making a second correction based on these candidate regions. Subsequently, Faster RCNN introduced a fully convolutional network as the RPN (Region Proposal Network), introducing the concept of Anchors for classification and bounding box regression, further improving the accuracy of object detection [20]. Following this, algorithms like FPN [38] and Mask RCNN [39] enriched the components of Faster RCNN, enhancing its performance by adding a branch to parallelly conduct pixel-level object instance segmentation. While two-stage object detectors offer high accuracy, their detection speed is relatively slow, and their complex detection model workflow limits their development on small terminal devices. Therefore, one-stage detectors, represented by YOLO, were introduced. YOLO restructures the detection problem, treating it as a regression problem and directly predicting image pixels as the target and its bounding box properties [40].

Subsequently, based on the improved YOLOv4 [41], techniques such as data augmentation, regularization methods, class label smoothing, CIoU-loss, CmBN (Cross mini-Batch Normalization), self-adversarial training, and cosine annealing learning rate scheduling were employed to enhance training. It utilized the “bag of freebies” approach, which only increases training time without affecting inference time. The recent YOLOv8, by referencing designs from algorithms like YOLOX [42], YOLOv6 [43], YOLOv7 [44] and PPYOLOE [45], offers a new SOTA (state-of-the-art) model. In terms of loss function design, it incorporates the Task Aligned Assigner positive sample allocation strategy and introduces Distribution Focal Loss to further reduce precision loss [46]. Despite YOLOv8’s outstanding performance in object detection tasks, it may face some challenges when dealing with complex scenarios like fully mechanized coal mine working faces. For instance, its large parameter model may not be effectively deployed for real-time object detection on terminal devices. Moreover, YOLOv8 typically requires extensive and diverse training data to achieve optimal performance. In certain specific domains or for certain target types, additional data may be needed for effective training.

In coal mine scene imagery, issues such as insufficient illumination and dim visibility are common challenges. Early researchers attempted to address these issues by resorting to methods like augmenting lighting fixtures or improving lighting layouts. However, such approaches often entail high costs, high energy consumption, and may not completely alleviate the problem of inadequate illumination. With the continual advancement of image-processing technologies, some researchers have turned to super-resolution reconstruction methods to enhance the details and clarity of digital images. For instance, single-image super-resolution (SISR) methods based on deep learning can produce realistically detailed reconstructions [47]. However, these methods suffer severe performance degradation in low-light conditions due to their neglect of the adverse effects of illumination. Cheng [48] pioneered an anti-illumination approach for SISR, termed Light-guide and Cross-fusion U-Net (LCUN), which simultaneously improves texture details and illumination for low-resolution images. Wu [49] proposed a Hybrid Super-Resolution (HYSR) framework that combines multi-image super-resolution (MISR) with single-image super-resolution to generate high-resolution images, thereby achieving superior spatial resolution.

In the field of coal mining production, the application of object detection technology continues to evolve. Yang et al. proposed the use of sensor-based technologies such as LiDAR and radar to detect coal mine obstacles, monitor geological structures, and ensure worker safety [50]. However, sensor performance is often influenced by environmental conditions, and underground extreme conditions may lead to a decline or failure in sensor performance. Additionally, the manufacturing and maintenance costs of high-performance sensors are relatively high [51].

In the realm of image-based object detection, Pan et al. introduced an improved fast recognition model based on YOLO-v3 for rapid identification of coal and gangue [52]. Zhang et al. presented a YOLOv4 algorithm based on deep learning for coal gangue detection. The detection algorithm with optimization methods showed higher accuracy, recall rate, and real-time performance compared to the SSD and Faster R-CNN detection algorithms [53]. Fan et al. proposed a coal particle morphology, particle size, liberation feature, and density separation process based on a CNN and an improved U-Net network model [54]. Wang et al. developed the Var-Con-Sin-GAN model and constructed a sample generation and feature transfer framework to address the issue of insufficient coal-rock image data [55]. The methods mentioned above have made some significant progress in the field of coal mine object detection, but there are still some challenges and areas for improvement. For instance, the application of some larger models in coal mine scenarios with limited computational resources may be restricted. Additionally, some models may face the challenge of reduced recognition accuracy in situations with inadequate lighting underground.

## 3. Materials and Methods

This paper introduces a lightweight object recognition algorithm tailored for fully mechanized coal mine working faces. The algorithm employs predefined Anchor Boxes to predict the sizes of target boxes, enhancing the speed and accuracy of object detection. Simultaneously, it optimizes the network structure through pruning operations based on the L1 norm, significantly reducing model computational operations and size while maintaining nearly undiminished object recognition accuracy. Consequently, CM-YOLOv8 (Coal Mining-YOLOv8) achieves streamlined and optimized deep learning models with reduced parameter count and computational complexity. It provides a more real-time, efficient, and adaptable image recognition solution for fully mechanized coal mine working faces, meeting the urgent demands of the mining industry for safety and production efficiency.

### 3.1. Predefined Anchor Box

This paper proposes a method for adaptive predefined anchor box generation using a genetic algorithm-based K-Means clustering analysis on the fully mechanized coal mine working face dataset to enhance object detection performance. The method begins by reading training set images and extracting the width (*w*), height (*h*), and bounding box sizes ($w_{bbox}$, $h_{bbox}$) for each image. Different ratios of *w* and *h* are proportionally scaled to a specified size while maintaining aspect ratios. Each image and its bounding box are then scaled proportionally, ensuring that the relative coordinates of the bounding box remain unchanged. The scaled width and height of the corresponding image are multiplied, converting the bounding box coordinates from relative to absolute coordinates. Bounding boxes with widths and heights of less than two pixels are filtered out because smaller objects may contribute more significantly to the training process. Next, K-means clustering is applied to the remaining bounding boxes to obtain *k* initial anchor boxes. This step helps initialize the genetic algorithm with diversified anchor box sizes. The genetic algorithm is then implemented to iteratively optimize the anchor box sizes. The algorithm involves randomly mutating the width and height of each anchor box and evaluating the fitness of the obtained set of anchor boxes using a custom fitness function. The mutation process is repeated up to 1000 times, adopting the new size if the mutation improves fitness; otherwise, the mutation is discarded. Finally, the anchor boxes are sorted based on their area, and the optimized anchor box sizes are returned for use in the object detection framework. The schematic diagram of the process is shown in Figure 1.

The K-Means algorithm used in this method first initializes *K* cluster centers $\left\{C_{1},C_{2},C_{3},\ldots ,C_{k}\right\},\;1<j\leq n$, and then calculates the Euclidean distance from each dataset sample $X=\left\{X_{1},X_{2},X_{3},\ldots ,X_{n}\right\}$ to each cluster center. The calculation formula is as follows:(1)$$dis(X_{i},C_{j})=\sqrt{\sum_{t=1}^{m}{\left(X_{it}-C_{jt}\right)}^{2}},$$
where $X_{it}$ represents the attribute *t* of the data sample *i*, and $C_{jt}$ represents the attribute *t* of the cluster center *j*. Subsequently, based on the distances, the dataset samples are partitioned into *K* clusters, and the mean is calculated for each cluster to update the cluster centers. This iterative process continues until the specified number of iterations is reached. The mean calculation is represented as follows:(2)$$Center_{k}=\frac{1}{\left|C_{k}\right|}\sum_{X_{i}\in C_{k}}X_{i}.$$

### 3.2. Pruning Based on L1 Norm

This paper addresses the issue of redundant network structures and computational complexity by proposing an L1 norm-based pruning method. The method divides the weight parameters of the convolutional layer to be pruned into *n* groups in the channel direction based on the parameter *group*, and calculates the L1 norm of weights on different channel numbers within each group. Pruning is then applied to the group’s weight values exceeding a certain threshold, while retaining and participating in retraining for the group’s weight values below the threshold. The weight parameters reaching the target accuracy after retraining are obtained, and the grouping and pruning operation is repeated until the network converges.

The calculation method for the intra-group channel L1 norm of each weight group in each convolutional layer to be pruned is shown in Equation (3). In the formula, *w* and *h* represent the maximum width and height of the current channel in the convolutional layer, *c* is the number of channels in the current convolutional kernel, *n* is the group after weight grouping in the convolutional layer to be pruned, *i* and *j* represent the current position of the weight value in the horizontal and vertical directions, $w_{i,j}$ is the weight value at the current position, *all* is the total number of weights in the convolutional layer, and ${\left\|w_{c}\right\|}_{1}$ is the L1 norm within channel *c*. Subsequently, pruning is determined based on comparing the L1 norm with the threshold within each group after segmentation.
(3)$${\left\|w_{c}\right\|}_{1}=\sum_{i=1,j=1}^{w,h}\left|w_{i,j}\right|.$$

The calculation method for the threshold $\theta_{n}$ within the group *n*, used to determine whether pruning should be applied, is shown in Equation (4). In the formula, $w_{j}$ represents the *j* weight in the convolutional layer, and *group* indicates the pruning parameter set for the current convolutional layer.
(4)$$\theta_{n}=\sum_{j=1}^{group}\left|w_{j}\right|/group.$$

The algorithmic mathematical model for channel-wise L1 norm-based pruning is illustrated in Algorithm 1. The time complexity of this algorithm is O(n). The input comprises the training data, pretrained model, hyperparameters such as group, various network training parameters, and pruning strategy. The output is the compressed model post-pruning. The pruning strategy is tailored to different network models; for instance, in this study, pruning operations are performed on modules like CBS in YOLOv8, aiming to minimize impact on recognition accuracy. Once the pruning module is determined, pruning thresholds $\theta_{n}$ are computed based on the L1 norms of grouped channel weights of convolutional layers and the hyperparameter group. Subsequently, pruning decisions are made for different groups based on these thresholds, with channels exceeding the threshold retained and others pruned. Finally, the pruned network undergoes retraining to restore recognition accuracy.
**Algorithm 1.** Channel L1 norm-based pruning algorithm**Input:** Training data: *X*, pre-trained weights: *W*, hyperparameter *group*,    pruning strategy:    neural network training parameters (such as learning rate, batch size, etc.),**Output:** Compressed weights  1: **for** pruning strategy **do**  2:   Obtain *θ*_*n*_ based on *W* and hyperparameter *group*  3:   **if**
*W*_*n*_ < *θ*_*n*_
**do**  4:     Channel L_1_ norm-based pruning on *W*_*n*_  5:   **else**
**do**  6:     Retrain the network  7: **end**
**for**

To provide a more illustrative description of how the L1 norm-based pruning method is applied to the CBS convolutional block, this paper designs the schematic diagram shown in Figure 2 for further clarification. Figure 2a demonstrates the pruning approach when the parameter *group* is set to 2. Each pair of channels in the convolutional layer is grouped, and pruning is applied to channels with values below the threshold, effectively multiplying them by a zero matrix. This approach has the advantage of significantly reducing computation in the output and subsequent network calculations, allowing the feature map size to remain unchanged without affecting further pruning and retraining. Channels with values above the threshold continue with the depth-wise separable convolution calculation. Figure 2b illustrates the pruning approach when the parameter *group* is set to 4, following a similar process where each group comprises four channels in the convolutional layer. Channels with values below the threshold undergo pruning, while those above the threshold continue with the depth-wise separable convolution calculation.

### 3.3. Network Structure

The lightweight pruning-enhanced algorithm network structure based on YOLOv8 for coal mining face is illustrated in Figure 3. The input size of this network structure remains at 640 × 640 × 3 and is divided into three main parts: Backbone, Neck, and Head. The PCBS in the network structure represents the module pruned based on L1 norm using depth-wise separable convolution, and its process schematic is depicted in Figure 3d. Depth-wise separable convolution is a specialized convolution operation in convolutional neural networks aimed at reducing the number of parameters and computational complexity while maintaining model expressiveness. This type of convolution operation is widely used in resource-constrained environments such as mobile devices and embedded systems to improve model lightweighting and operational efficiency. However, the conventional YOLOv8 network structure involves numerous CBS, leading to parameter redundancy and repetitive computations. Hence, this paper performs L1-norm-based pruning on the convolutional part of CBS, reducing parameter and computation overhead while striving to maintain recognition accuracy, simultaneously applying PCBS to each depth-wise separable convolution of the YOLOv8.

Meanwhile, YOLOv8 replaces the commonly used C3 module in the YOLO series algorithms with the C2f module, introducing more skip connections and additional Split operations [56]. The C3 module inserts a CSP (Cross Stage Partial) connection between each branch in the CSP structure, dividing the feature map into two parts. One part undergoes multiple layers of convolution before merging with the other part, aiding in information integration at different levels. The C2f module is designed based on the ideas of the C3 module and ELAN, as shown in Figure 3e. This design allows YOLOv8 to obtain richer gradient flow information while ensuring the light weight of the network. Figure 3f illustrates the detailed structure of the SPPF module. SPP (Spatial Pyramid Pooling) is commonly used in deep learning for spatial pyramid pooling methods [57]. This module is typically used to handle variations in input sizes to adapt to objects or scenes of different sizes. The SPPF module further optimizes the operation sequence and size of Maxpooling based on SPP, maintaining consistent sizes while accelerating computation speed. It addresses the problem of convolutional neural networks extracting redundant features related to the image, significantly improving the speed of generating candidate boxes.

## 4. Results

### 4.1. Experiment Introduction

This section begins by introducing the dataset used in the experimental methodology, followed by descriptions of the experimental environment and training strategies. Finally, evaluation metrics related to the experimental results are presented.

#### 4.1.1. Dataset

The dataset utilized in this study is the Underground Longwall Mining Face (DsLMF+) image dataset [58], comprising 138,004 images annotated with six classes: coal miner, hydraulic support guard plate, large coal, mine safety helmet, miner behaviors, and towline. The dataset incorporates diverse angles, scenes, and tasks, providing a comprehensive representation with varied object categories (monotonous and diverse), object quantities (few and abundant), and object distributions (sparse and dense). Some representative images from the dataset are illustrated in Figure 4. All labels in the dataset are openly available in both YOLO and COCO formats, and domain experts in the mining field have assessed the dataset’s utility and accuracy.

Through the genetic-algorithm-based K-Means clustering analysis on the coal mining face dataset, this study discovered that to achieve better recovery of pruning accuracy, the predefined aspect ratios for various target anchors should be as follows: Coal miner: [2.6, 2.9], Hydraulic support guard plate: [1.9, 2.3], Large coal: [0.8, 1.3], Mine safety helmet: [0.9, 1.2], Miner behaviors: [2.1, 2.5], and Towline: [4.1, 4.5].

#### 4.1.2. Experimental Environment and Training Strategies

The experimental setup for the CM-YOLOv8 algorithm in this study is composed of the components listed in Table 1. This includes the hardware platform and the deep learning framework employed in the experiments.

The YOLOv8m model was employed as the backbone network for improvement training in this study. This model follows all the experimental ideas of the YOLOv8 series, with the only modification being the scaling of the network’s width and depth. The crucial parameter settings for the training process are outlined in Table 2.

Following the partition rules of the DsLMF+ coal mine dataset, the dataset was divided into a training set (110,403 images), a test set (16,800 images), and a validation set (10,801 images).

#### 4.1.3. Evaluation Indicators

To provide a detailed and accurate description of the excellent performance of the proposed CM-YOLOv8 improved model in this study, various evaluation metrics were introduced, including precision, recall, mAP0.5, mAP0.5:0.95, model parameter count, and model size, where precision is how many of the samples predicted by the model to be in the positive category are truly in the positive category. It is calculated as shown in Equation (5):(5)$$Precision=\frac{TP}{TP+FP}$$
where *TP* (True Positives) represents the number of samples correctly predicted as the positive class by the model, and *FP* (False Positives) represents the number of samples incorrectly predicted as the positive class by the model.

Recall refers to how many actual positive class samples are correctly predicted as the positive class by the model. The calculation is shown in Equation (6):(6)$$Recall=\frac{TP}{TP+FN}$$
where *FN* (False Negatives) represents the number of samples incorrectly predicted as the negative class by the model.

*AP* (Average Precision) is equal to the area under the Precision–Recall curve, and the calculation Equation is shown in Equation (7):(7)$$AP=\int_{0}^{1}Precision(Recall)d(Recall)$$
mAP (Mean Average Precision) is the weighted average of the *AP* values for all sample categories used to measure the detection performance of the model across all categories. The equation is shown in Equation (8):(8)$$mAP=\frac{1}{N}\sum_{i=1}^{N}AP_{i}$$
where ${AP}_{i}$ represents the *AP* value for the category with index *i*, and *N* is the number of categories in the training dataset. Additionally, mAP0.5 is the mean average precision at an IoU (Intersection over Union) threshold of 0.5. In object detection, IoU is used to measure the overlap between the predicted bounding box and the ground truth bounding box [59]. mAP0.5:0.95 is an extension of mAP over a broader IoU range, similar to mAP0.5 but considering a wider IoU range from 0.5 to 0.95.

### 4.2. Experiment Results

This paper takes the DsLMF+ dataset as an example and conducts thorough validation on the coal mining dataset. To ensure that the test results of different network models are not influenced by factors other than model differences, consistent parameter settings are maintained for all network models during the experiments.

#### 4.2.1. Quantitative Comparison of Different Models

This paper compares the performance of YOLOv7 [44], DETA [60], ViT-Adapter [61], YOLOv8 [27], and the proposed CM-YOLOv8 algorithm on the DsLMF+ dataset, as shown in Table 3. From the table, it can be observed that compared to YOLOv7, YOLOv8 shows improved recognition accuracy due to its deeper network architecture. Additionally, YOLOv8 enhances detection speed by replacing the traditional C3 module with the C2f module, partially addressing its speed limitation. DETA, built on the Deformable DETR two-stage architecture, adopts a one-to-many matching strategy with a traditional IOU-based matching policy in CNN. Although DETA performs well overall, it is slightly inferior to YOLOv8. ViT (Vision Transformer), a vision processing model based on the Transformer architecture, shows relatively mediocre performance on the coal mining dataset. This could be attributed to ViT’s emphasis on global information over local features, resulting in poorer recognition of details and local structures in the coal mining face scenario. The proposed CM-YOLOv8 achieves a slight decrease in average recognition accuracy compared to the YOLOv8m model (0.1%) while significantly reducing computational and parameter overhead. The average time to recognize an image in this model is 234 ms with an input image of 640 × 640 and target detection of the six targets in the paper, which is 43.7 GFLOPs when the parameter group is set to 4, and 23.1 GFLOPs when the parameter group is set to 2, compared to 78.9 GFLOPs for the parameter number of the traditional YOLOv8. 78.9 GFLOPs. From the table, it is evident that in the recognition accuracy of Coal miners, Miners’ behaviors, and Towline targets, the proposed algorithm maintains accuracy even after removing network redundancy.

#### 4.2.2. Comparison of Loss Function Changes during Training for Different Kinds of Targets

This paper initially sets the training epochs to 300. However, after exceeding 100 epochs, the change in mAP becomes minimal and lacks visual value. Therefore, this paper illustrates the mAP variation curve for the first 100 epochs. As shown in Figure 5, the convergence of the training process for six types of target objects, including Coal miners, Mine safety helmet, Hydraulic support guard plate, Large Coal, Miners’ behaviors, and Towline, is depicted. From the figure, it can be observed that Towline achieves the highest mAP0.5 recognition accuracy among all target objects, reaching 99.6%. In contrast, Large Coal has the lowest mAP0.5 recognition accuracy, which is 87.1%. This may be attributed to the significant diversity and complexity in the shape, color, and texture of large coal blocks, making feature extraction and matching more challenging. On the other hand, cable groove appearances are relatively consistent, making it easier to recognize and extract features. Each type of target generally converges before reaching 20 epochs, but Miners’ behaviors require a relatively longer convergence time. This is because the appearance features of individuals are relatively easy to extract, typically involving static features such as faces and bodies. In contrast, behavioral features may include dynamic movements, poses, and other information, making their extraction more challenging and requiring a longer convergence time.

#### 4.2.3. Comparison of Computational and Model Parametric Quantities after Lightweight Pruning

This paper addresses the practical hardware deployment requirements for scenarios such as coal mining comprehensive mining faces and proposes a lightweight pruning method based on L1 norm. The comparative results of various methods, including recognition accuracy, parameter quantity, and computational load on the coal mining dataset, are presented in Table 4, where GFLOPs represent Giga Floating-point Operations Per Second. From the table, it can be observed that early target detection algorithms, such as YOLOv3 [62] and HTC [63], are not conducive to deployment on small-terminal devices in coal mines due to their complex model structure and large parameter quantities, and their detection accuracy is relatively low. Although YOLOv7 achieves extreme compression of parameter and computational loads, its detection accuracy also significantly decreases, making it challenging to meet the safety requirements of actual coal mining production. Compared to other algorithms, YOLOv7 and YOLOv8m achieve a reduction in parameter quantity and computational load while ensuring high recognition accuracy. However, in the context of limited computing resources in coal mine production, these models are still challenging to deploy on small-terminal devices on a large scale. Moreover, both models adopt a three-scale detection network structure, which cannot meet the detection requirements for high-proportion small targets in the coal mining production scene, resulting in deficiencies in detection accuracy. In contrast, CM-YOLOv8, proposed in this paper, significantly compresses the required parameter quantity and computational load for model operation while ensuring that the recognition accuracy does not decrease or decreases minimally. Taking the network model with parameter setting *group = 2* as an example, compared to YOLOv8m, the proposed model reduces the parameter quantity by 39.8% and the computational load by 44.6% while only sacrificing 0.1% in recognition accuracy. This enables the proposed network model to provide technical support for real-time deployment on small devices in coal mining comprehensive mining face scenarios.

In tasks related to coal mine scene object detection, some models with more parameters and computational complexity may perform poorly. This could be attributed to the larger models inadequately learning feature representations. Despite their increased complexity due to the augmented parameter count, larger models do not necessarily excel in learning effective feature representations from the data. If a model fails to effectively capture features relevant to object recognition, its performance remains subpar regardless of its scale. Moreover, the increased depth and complexity in larger models may exacerbate issues like gradient vanishing or explosion during backpropagation, leading to unstable training and difficulty in converging to satisfactory accuracy. Furthermore, larger models may inadvertently introduce more noise or suffer from information loss during information propagation, thereby diminishing overall performance.

#### 4.2.4. Comparison of Recognition Results for Visualization of Different Kinds of Targets

In order to visually and conveniently depict the detection effect of the model proposed in this paper, this paper conducted a comparison experiment using the confusion matrix with the YOLOv8m model, and the comparison results are shown in Figure 6. The rows and columns of the confusion matrix represent the true and predicted categories, respectively, and the values in the diagonal region represent the proportion of correctly predicted categories, while the values in the other regions represent the proportion of incorrectly predicted categories. As can be seen from Figure 6, the color of the diagonal region of the confusion matrix in CM-YOLOv8 is darker than that of YOLOv8m, which indicates that the ability of the model in this paper to correctly predict object categories is enhanced. Meanwhile, the miss detection rate of CM-YOLOv8 compared with YOLOv8m is effectively reduced, especially in terms of the two similar categories of coal miners and coal miners’ behavior.

The performance of the proposed CM-YOLOv8 algorithm in real coal mining comprehensive mining face scenarios is depicted in Figure 7. The figure illustrates the visual results of the algorithm in recognizing targets such as Coal miners, Mine safety helmet, Hydraulic support guard plate, Large Coal, Miners’ behaviors, and Towline. The main distinction between “coal miners” and “miner behaviors” lies in the different spatial locations. If coal mine workers are found standing within the hazardous area on the inner side of the hydraulic support, typically, such a scenario would be classified as “miners’ risky behavior”. In contrast, the identification of coal miners does not involve surrounding environmental information but focuses solely on recognizing individuals belonging to the category of coal miners. It is evident that the model proposed in this paper performs well on the coal mining dataset, successfully identifying targets in coal mine images. The accurate identification boxes and confidence scores enable high-precision localization of target objects.

Compared to traditional YOLO series algorithms, the proposed model benefits from fixed-size bounding boxes, making it more precise in recognizing specific target objects. This avoids the decrease in the model’s recognition performance for targets of different scales in the image due to factors such as distance, angle, or image resolution. Thanks to the multi-scale design adopted by YOLOv8 series algorithms, the proposed algorithm can detect targets at different scales simultaneously. This allows the algorithm to improve its recognition capability for small or low-light targets in dark conditions underground in coal mines.

Moreover, the lightweight YOLOv8 reduces the computational requirements of the model through pruning, enhancing the model’s inference speed. This is especially important for scenarios such as coal mining face that require real-time monitoring, allowing for more accurate and timely identification of targets and the adoption of appropriate measures. Additionally, using lightweight models on embedded or mobile devices can reduce power consumption, extending the device’s battery life—a critical aspect for coal mine monitoring systems that need to operate for extended periods.

## 5. Conclusions

This study addresses challenges stemming from inadequate image data, high model complexity, and limited computational resources in coal mining face scenarios. It introduces a lightweight object detection algorithm specifically tailored for coal mining face applications. This algorithm improves detection performance for various targets within a coal mine by generating adaptive predefined anchor boxes suitable for the coal mining face dataset. Additionally, a pruning method based on the L1 norm is devised to significantly reduce model computation and parameter complexity while preserving accuracy. Experimental results obtained with CM-YOLOv8 on the coal mining dataset are compared with those of other algorithms, demonstrating its efficiency and practicality in coal mining scenarios. Nevertheless, there remains room for further enhancement in this research endeavor. Due to the strict confidentiality surrounding coal mining datasets, the availability of open-source datasets is limited, posing challenges for large-scale model training. Furthermore, optimizing the deployment of this algorithm on computing devices with constrained resources and streamlining hardware deployment processes will be pivotal for future improvements. Subsequent research could delve into employing transfer learning techniques to leverage models trained in different domains for enhancing object detection performance in coal mining scenarios. Additionally, exploring weakly supervised learning methods may help reduce reliance on labeled data.

## Figures and Tables

**Figure 1 sensors-24-01866-f001:**
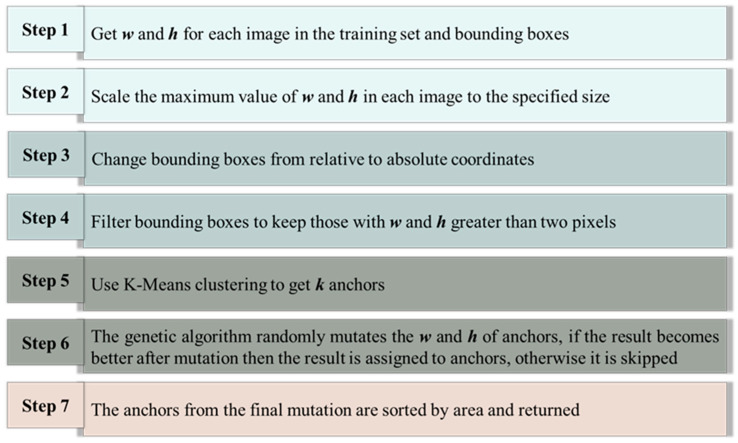
The steps of predefined Anchor Box generation method based on genetic algorithm.

**Figure 2 sensors-24-01866-f002:**
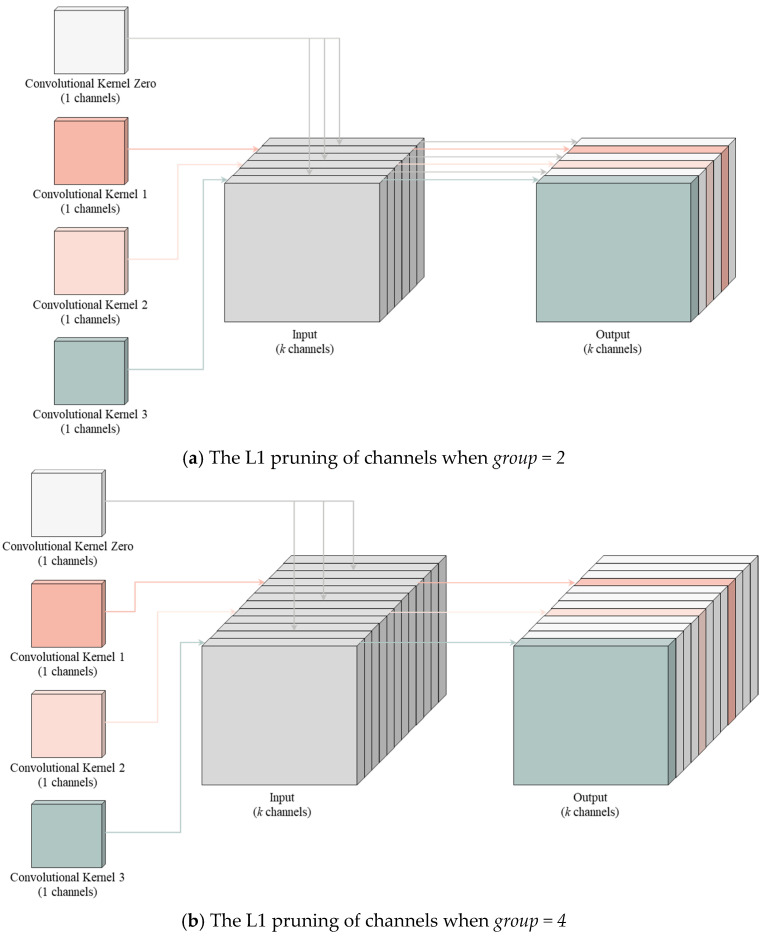
Channel L1 norm-based pruning algorithm. (**a**) The L1 pruning of channels when *group = 2*. In the figure, “Convolutional Kernel 1”, “Convolutional Kernel 2”, and “Convolutional Kernel 3” represent the retained convolutional kernels, while “Convolutional Kernel Zero” denotes the zero-mask matrix. They are multiplied with the input, resulting in the output. Within the output, only one channel of information is retained for every two channels. (**b**) The L1 pruning of channels when *group = 4*. In the figure, “Convolutional Kernel 1”, “Convolutional Kernel 2”, and “Convolutional Kernel 3” represent the retained convolutional kernels, while “Convolutional Kernel Zero” denotes the zero-mask matrix. They are multiplied with the input, resulting in the output. Within the output, only one channel of information is retained for every four channels.

**Figure 3 sensors-24-01866-f003:**
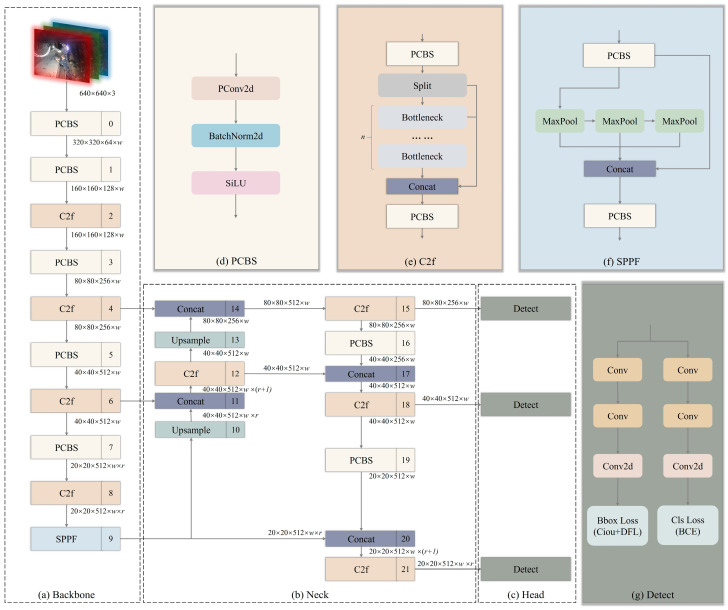
L1-paradigm-based operation of pruned CBS convolutional block.

**Figure 4 sensors-24-01866-f004:**
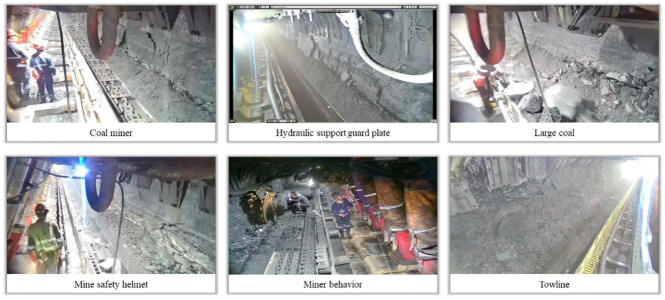
Samples of the DsLMF+ dataset.

**Figure 5 sensors-24-01866-f005:**
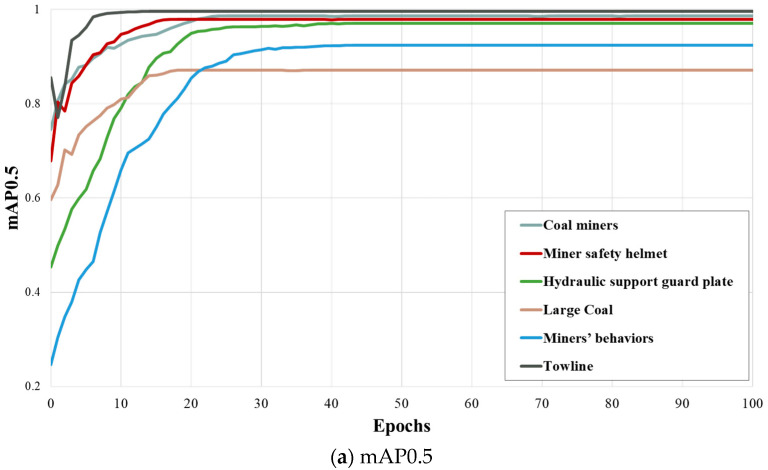
Comparison of mAP changes during training for different objects.

**Figure 6 sensors-24-01866-f006:**
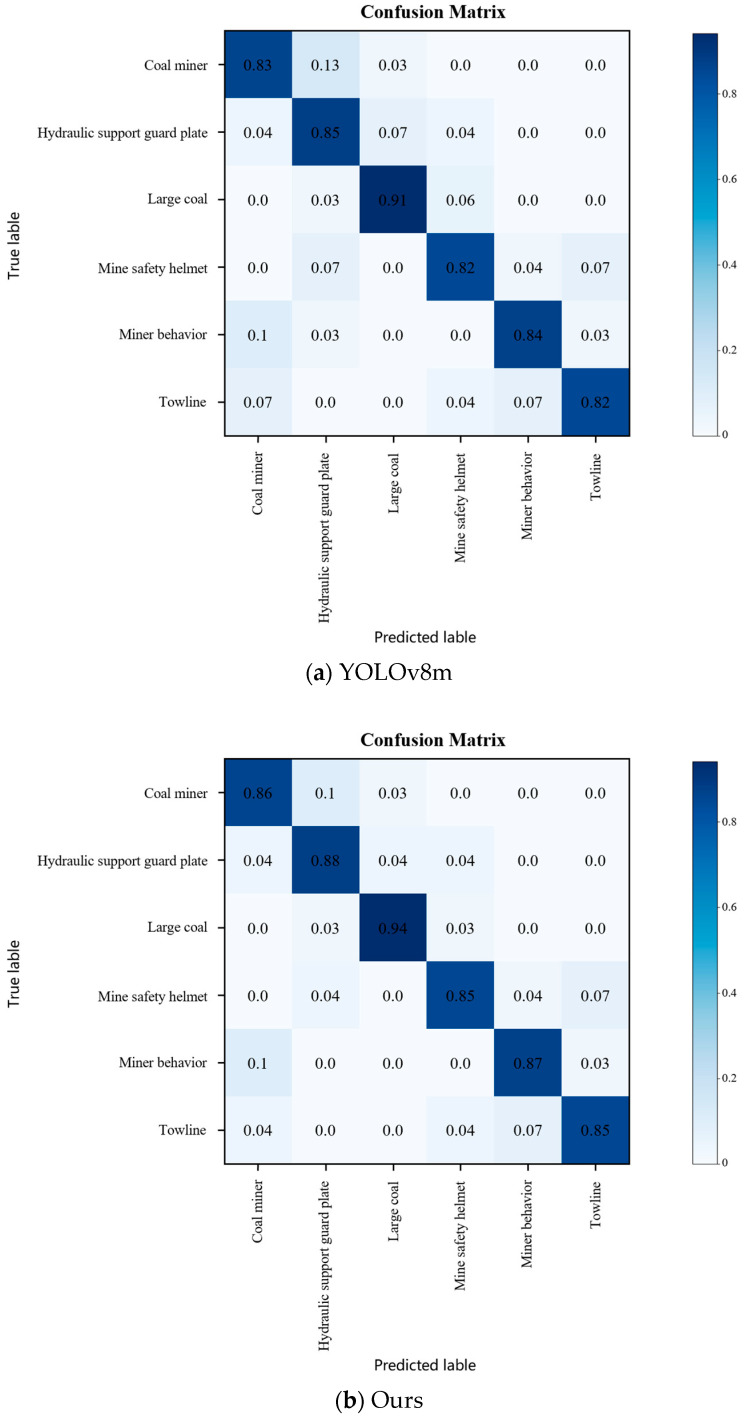
Confusion matrix plot of different models.

**Figure 7 sensors-24-01866-f007:**
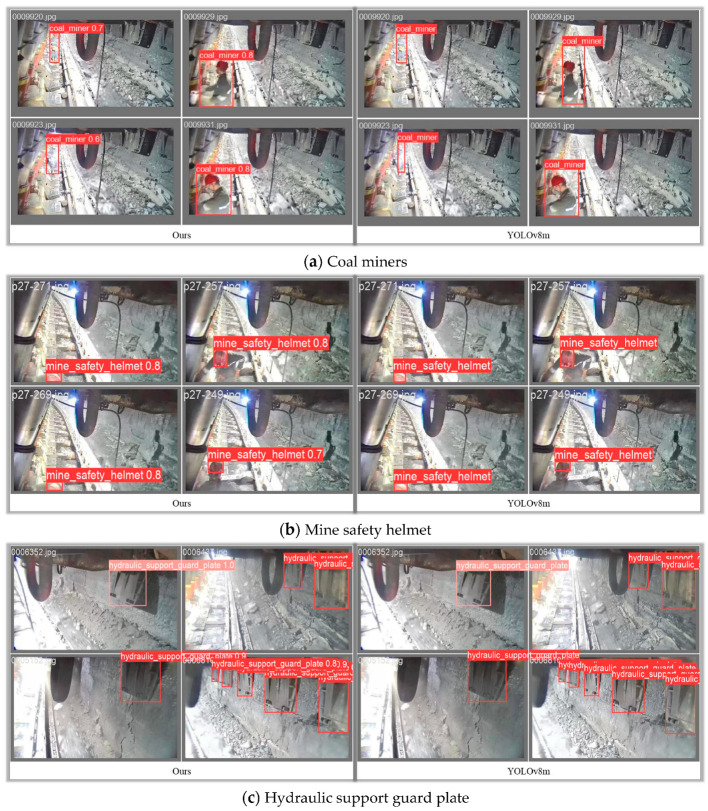
Comparison of the effect of visualization identification of various types of targets in coal mine working face.

**Table 1 sensors-24-01866-t001:** Composition of experimental algorithm configuration workstation.

Parameters	Configuration
CPU	Intel Core i9-13900HX (Intel Corporation, Shanghai, China) (2.2 GHz, 6 cores, 12 threads)
RAM	16 GB DDR4 2400 MHz
GPU	NVIDIA GeForce GTX 4070 Ti × 2 (NVIDIA Corporation, Beijing, China)
GPU memory size	16 GB
cuDNN	8.9.6
CUDA	12.1
Deep learning framework	python 3.8.18 + pytorch 2.1.2

**Table 2 sensors-24-01866-t002:** Important parameter settings during experimental training.

Parameters	Value
Epochs	300
Initial learning rate	1 × 10^−2^
Final learning rate	1 × 10^−4^
Momentum	0.937
Weight-Decay	5 × 10^−4^
Batch size	4
Mosaic	1.0
α(Wise-IoU)	1.9
δ(Wise-IoU)	3
Input image size	640 × 640
Number of images	138,004
Optimizer	SGD
Close Mosaic	Last 10 epochs

**Table 3 sensors-24-01866-t003:** Comparison of quantitative experimental accuracy of different models.

Method	YOLOv7				DETA				ViT-Adapter-L				YOLOv8m				Ours			
Metrics	Precision	Recall	mAP 0.5	mAP 0.5:0.95	Precision	Recall	mAP 0.5	mAP 0.5:0.95	Precision	Recall	mAP 0.5	mAP 0.5:0.95	Precision	Recall	mAP 0.5	mAP 0.5:0.95	Precision	Recall	mAP 0.5	mAP 0.5:0.95
Coal miners	0.965	0.968	0.986	0.773	0.967	0.970	0.976	0.684	0.961	0.965	0.966	0.702	0.968	0.971	0.988	0.775	0.968	0.970	0.986	0.774
Mine safety helmet	0.942	0.958	0.976	0.679	0.948	0.965	0.960	0.601	0.945	0.951	0.961	0.624	0.946	0.962	0.980	0.680	0.943	0.959	0.979	0.680
Hydraulic support guard plate	0.972	0.927	0.978	0.813	0.971	0.932	0.958	0.762	0.963	0.928	0.963	0.753	0.978	0.927	0.974	0.817	0.972	0.925	0.971	0.812
Large Coal	0.814	0.776	0.868	0.572	0.820	0.771	0.815	0.549	0.811	0.776	0.854	0.532	0.815	0.780	0.873	0.580	0.814	0.786	0.871	0.574
Miners’ behaviors	0.880	0.880	0.913	0.752	0.884	0.886	0.914	0.718	0.879	0.874	0.928	0.714	0.882	0.882	0.926	0.754	0.883	0.879	0.924	0.754
Towline	0.995	0.997	0.997	0.916	0.996	0.998	0.989	0.915	0.995	0.992	0.989	0.871	0.997	0.997	0.996	0.920	0.997	0.996	0.996	0.919
Average	0.928	0.918	0.953	0.751	0.931	0.920	0.935	0.705	0.926	0.914	0.944	0.699	0.931	0.920	0.956	0.754	0.930	0.920	0.955	0.752

**Table 4 sensors-24-01866-t004:** Comparison of Computational and Model Parametric Quantities after Lightweight Pruning.

Method	mAP:0.5(%)	Params(M)	GFLOPs
YOLOv3 [62]	85.9	61.5	193.9
YOLOv5m [64]	87.7	21.8	39.4
YOLOv7 [44]	95.3	36.5	103.5
YOLOv7-tiny [65]	78.4	6.0	13.1
YOLOv8m [27]	95.6	25.9	78.9
Faster-RCNN [20]	86.2	40.2	207.3
DETR [66]	79.4	41.9	225.7
DAB-DETR [67]	90.3	44.8	256.1
HTC [63]	82.3	80.5	441.3
Ours (*group = 2*)	95.5	15.6	43.7
Ours (*group = 4*)	94.1	9.7	23.1

## Data Availability

Data are contained within the article.

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
