# Peer review of "CM-YOLOv8: Lightweight YOLO for Coal Mine Fully Mechanized Mining Face"

_sensors, 2024, doi:10.3390/s24061866_

Round 1
Reviewer 1 Report
Comments and Suggestions for Authors
1. What is the time complexity of the proposed model?
2. A non-parametric statistical analysis should be carried out to compare the performance of the proposed model with existing models
3. The conclusion section should be improved
Author Response
Please see the attachment, thank you!

Reviewer 2 Report
Comments and Suggestions for Authors
The original paper is provided for publication with the topic that could be interesting for the specific group of readers. Authors have presented the algorithm of adaptive predefined anchor boxes tailored to the coal mining face dataset to enhance the detection performance of various targets. As an added value of the article, the authors single out the improved Yolov8 operation algorithm, which was supplemented with pruning method based on the L1 norm. According to the authors and presented results, it can be said that the yolov8 model parameter volume was compressed without losing the classification accuracy. It can also be stated that the article itself has all the necessary components for a scientific article.
On the other hand, the article also has obvious questions that should be answered.
1. The experiment was conducted on a powerful computer with an Intel Core i9 motherboard, 16 GB of RAM, and a powerful GPU, although the study emphasizes that the solution is designed for a resource-constrained embedded computer. Therefore, I intuitively kept waiting for a description of the resource limited embedded computer. I was a little lost at first when I didn't find it. It would be useful and meaningful to implement experiments in real conditions on a device with appropriate computing power.
– In this case, I would suggest either repeating the experiments, specifying a particular device with limited resources: Jetson nano, Raspbery PI or some other. At least, describe the devices to which you are aligning to. Because the device and resources can influence the algorithm itself and whether the device can handle the task.
– Another option is to emphasize that, in this case, it is a theoretical study focusing on limited resources, but the experiment will be conducted on a powerful computer, and only four metrics of the proposed solution will be provided.
2. It is written “Simultaneously, a pruning method based on the L1 norm is designed, significantly compressing the model's computation and parameter volume without compromising accuracy”.
What parameter in the results describes reduced computation resource? Again it would be meaningful to make classification on embedded computer with limited resource.
3. Python 3.8.13 + PyTorch 2.1.2 software packages were used in the study. The article mentions that the network structure was optimized using pruning based on L1 norm. However, it is not specified how this was implemented. Therefore, it remains unclear what specific changes were made to the YOLOv8m algorithm. Therefore, it is challenging to understand the level of effort the authors had to invest themselves. Did you create your specific L1 norm filter, or did you use the PyTorch implemented pruning method with the L1 norm argument, such as prune.l1_unstructured? If you developed your solution, a mathematical model should be provided. If you used an existing implementation, you still need to explain how it works mathematically to add scientific soundness to the article.
4. From the description of the study, it still remains entirely unclear to me how the class "miner's behavior" is classified and what the difference is then from the class "coal miner." In Figure 6a and e, the miner is standing, squatting and walking, and in one place, it is classified under the "coal miner" class, while elsewhere under the "miner's behavior" class. This aspect needs to be elucidated in more detail; otherwise, the classification algorithm remains unclear. To theoretically determine the class "miner's behavior," a sequence of frames would likely be necessary. For all other classes, individual frames would be sufficient. However, when classifying with YOLOv8, does it use a sequence of frames, evaluating a buffer of frames of some size, considering the history at different time points?
5. Summarizing, the scope of the article is extensive and could be more concise. In such a large volume, I missed essential information about the difficulties/uncertainties the authors encountered. It is not clear where the scientific problem lies, and it seems to focus more on an engineering solution with research results. It is necessary to highlight the scientific part of the article by identifying scientific uncertainties.
For example, in Figure 3, I strongly feel that the description is lacking in highlighting and specifying what exactly was implemented by the authors. It would be beneficial to indicate which parts, perhaps even using bold lines in the figure, were implemented by the authors. As mentioned in my previous comments, providing more precise details in the text about the means by which it was implemented would enhance clarity.
7. The article includes a list of 81 references. It is not a review article, the number of references seems excessive. It might not be crucial, but it's worth noting that 25% of the references are relatively old and, in my opinion, unnecessary.
6. There are also some small errors:
Table 4. The abbreviation GFLOPs is completely unexplained...
In Figure 2, a more detailed explanation is needed. Perhaps additional labels directly on the figure could be provided, as it remains unclear what fundamentally changed in parts a and b on the right side of the figure. The text below the image does not reveal this.
Formulas 2, 3, and 4 should end with dot. After Formula 1, a comma needs to be inserted, and the naming of variables in the formula should continue in line 200 with a lowercase letter without indentation from the left margin.
Comments on the Quality of English LanguageEnglish language is fine, only small corrections are required.
Author Response
Please see the attachment, thank you!

Reviewer 3 Report
Comments and Suggestions for Authors
To address the challenges of insufficient image data, high model complexity, and limited computational resources in coal mining face scenarios, the author proposes a light-weight object detection algorithm tailored for coal mining face applications. The innovation mainly includes the network pruning method and the adaptive predefined anchor boxes to achieve lightweight and high accuracy.This work has achieved a good balance in performance and lightweight, and it has a certain innovative and practical value. But I think there are still some minor suggestions to improve the current version, as follows:
(1) From the examples in the displayed dataset, it can be seen that coal mine images suffer from serious lighting problems. It is recommended to introduce some research on this lighting problem in the background of research status., such as:
Light-guided and Cross-fusion U-Net for Anti-illumination Image Super-resolution[J]. IEEE Transactions on Circuits and Systems for Video Technology, 2022, 32(12): 8436-8449.
(2) The predefined anchor box shows very good performance due to the priori on the dataset, but does this fixed anchor box negatively affect its performance on new data. Will it lead to poor robustness of the model?
(3) In the experiment, are the compared methods trained on the same coal mine dataset and setting? In addition, some models with more parameters and more calculations perform poorly. Please add the possible reasons for this phenomenon.
(4) Some figures are ambiguous, so it is recommended to replace them with vector images or higher resolution images.
Comments on the Quality of English LanguageIt is suggested to give the full names of some abbreviations where they first appear.
Author Response
Please see the attachment, thank you!

Round 2
Reviewer 2 Report
Comments and Suggestions for Authors
The authors has taken into account all my comments and now the idea, experiment, results looks clear to me. Therefore I recommend the paper for publication.
Author Response
N